# A Disentangled Recognition and Nonlinear Dynamics Model for Unsupervised Learning

**Marco Fraccaro**[†*]     **Simon Kamronn** [†*]     **Ulrich Paquet**[‡]     **Ole Winther**[†]

[†] Technical University of Denmark

[‡] DeepMind

## Abstract

This paper takes a step towards temporal reasoning in a dynamically changing video, not in the pixel space that constitutes its frames, but in a latent space that describes the non-linear dynamics of the objects in its world. We introduce the Kalman variational auto-encoder, a framework for unsupervised learning of sequential data that disentangles two latent representations: an object's representation, coming from a recognition model, and a latent state describing its dynamics. As a result, the evolution of the world can be imagined and missing data imputed, both without the need to generate high dimensional frames at each time step. The model is trained end-to-end on videos of a variety of simulated physical systems, and outperforms competing methods in generative and missing data imputation tasks.

## 1 Introduction

From the earliest stages of childhood, humans learn to represent high-dimensional sensory input to make temporal predictions. From the visual image of a moving tennis ball, we can imagine its trajectory, and prepare ourselves in advance to catch it. Although the act of recognising the tennis ball is seemingly independent of our intuition of Newtonian dynamics [31], very little of this assumption has yet been captured in the end-to-end models that presently mark the path towards artificial general intelligence. Instead of basing inference on any abstract grasp of dynamics that is learned from experience, current successes are autoregressive: to imagine the tennis ball's trajectory, one forward-generates a frame-by-frame rendering of the full sensory input [5, 7, 23, 24, 29, 30].

To disentangle two latent representations, an object's, and that of its dynamics, this paper introduces *Kalman variational auto-encoders (KVAEs)*, a model that separates an intuition of dynamics from an object recognition network (section 3). At each time step $t$, a variational auto-encoder [18, 25] compresses high-dimensional visual stimuli $\mathbf{x}_t$ into latent encodings $\mathbf{a}_t$. The temporal dynamics in the learned $\mathbf{a}_t$-manifold are modelled with a linear Gaussian state space model that is adapted to handle complex dynamics (despite the linear relations among its states $\mathbf{z}_t$). The parameters of the state space model are adapted at each time step, and non-linearly depend on past $\mathbf{a}_t$'s via a recurrent neural network. Exact posterior inference for the linear Gaussian state space model can be preformed with the Kalman filtering and smoothing algorithms, and is used for imputing missing data, for instance when we imagine the trajectory of a bouncing ball after observing it in initial and final video frames (section 4). The separation between recognition and dynamics model allows for missing data imputation to be done via a combination of the latent states $\mathbf{z}_t$ of the model and its encodings $\mathbf{a}_t$ only, without having to forward-sample high-dimensional images $\mathbf{x}_t$ in an autoregressive way. KVAEs are tested on videos of a variety of simulated physical systems in section 5: from raw visual stimuli, it "end-to-end" learns the interplay between the recognition and dynamics components. As KVAEs can do smoothing, they outperform an array of methods in generative and missing data imputation tasks (section 5).

---

[*]Equal contribution.

## 2  Background

**Linear Gaussian state space models.**  Linear Gaussian state space models (LGSSMs) are widely used to model sequences of vectors $\mathbf{a} = \mathbf{a}_{1:T} = [\mathbf{a}_1, .., \mathbf{a}_T]$. LGSSMs model temporal correlations through a first-order Markov process on latent states $\mathbf{z} = [\mathbf{z}_1, .., \mathbf{z}_T]$, which are potentially further controlled with external inputs $\mathbf{u} = [\mathbf{u}_1, .., \mathbf{u}_T]$, through the Gaussian distributions

$$p_{\gamma_t}(\mathbf{z}_t|\mathbf{z}_{t-1}, \mathbf{u}_t) = \mathcal{N}(\mathbf{z}_t; \mathbf{A}_t\mathbf{z}_{t-1} + \mathbf{B}_t\mathbf{u}_t, \mathbf{Q}) , \qquad p_{\gamma_t}(\mathbf{a}_t|\mathbf{z}_t) = \mathcal{N}(\mathbf{a}_t; \mathbf{C}_t\mathbf{z}_t, \mathbf{R}) . \quad (1)$$

Matrices $\gamma_t = [\mathbf{A}_t, \mathbf{B}_t, \mathbf{C}_t]$ are the state transition, control and emission matrices at time $t$. $\mathbf{Q}$ and $\mathbf{R}$ are the covariance matrices of the process and measurement noise respectively. With a starting state $\mathbf{z}_1 \sim \mathcal{N}(\mathbf{z}_1; \mathbf{0}, \boldsymbol{\Sigma})$, the joint probability distribution of the LGSSM is given by

$$p_\gamma(\mathbf{a}, \mathbf{z}|\mathbf{u}) = p_\gamma(\mathbf{a}|\mathbf{z})\, p_\gamma(\mathbf{z}|\mathbf{u}) = \prod_{t=1}^{T} p_{\gamma_t}(\mathbf{a}_t|\mathbf{z}_t) \cdot p(\mathbf{z}_1) \prod_{t=2}^{T} p_{\gamma_t}(\mathbf{z}_t|\mathbf{z}_{t-1}, \mathbf{u}_t) , \qquad (2)$$

where $\gamma = [\gamma_1, .., \gamma_T]$. LGSSMs have very appealing properties that we wish to exploit: the filtered and smoothed posteriors $p(\mathbf{z}_t|\mathbf{a}_{1:t}, \mathbf{u}_{1:t})$ and $p(\mathbf{z}_t|\mathbf{a}, \mathbf{u})$ can be computed exactly with the classical Kalman filter and smoother algorithms, and provide a natural way to handle missing data.

**Variational auto-encoders.**  A variational auto-encoder (VAE) [18, 25] defines a deep generative model $p_\theta(\mathbf{x}_t, \mathbf{a}_t) = p_\theta(\mathbf{x}_t|\mathbf{a}_t)p(\mathbf{a}_t)$ for data $\mathbf{x}_t$ by introducing a latent encoding $\mathbf{a}_t$. Given a likelihood $p_\theta(\mathbf{x}_t|\mathbf{a}_t)$ and a typically Gaussian prior $p(\mathbf{a}_t)$, the posterior $p_\theta(\mathbf{a}_t|\mathbf{x}_t)$ represents a stochastic map from $\mathbf{x}_t$ to $\mathbf{a}_t$'s manifold. As this posterior is commonly analytically intractable, VAEs approximate it with a variational distribution $q_\phi(\mathbf{a}_t|\mathbf{x}_t)$ that is parameterized by $\phi$. The approximation $q_\phi$ is commonly called the recognition, encoding, or inference network.

## 3  Kalman Variational Auto-Encoders

The useful information that describes the movement and interplay of objects in a video typically lies in a manifold that has a smaller dimension than the number of pixels in each frame. In a video of a ball bouncing in a box, like Atari's game Pong, one could define a one-to-one mapping from each of the high-dimensional frames $\mathbf{x} = [\mathbf{x}_1, .., \mathbf{x}_T]$ into a two-dimensional latent space that represents the position of the ball on the screen. If the position was known for consecutive time steps, for a set of videos, we could learn the temporal dynamics that govern the environment. From a few new positions one might then infer where the ball will be on the screen in the future, and then imagine the environment with the ball in that position.

The *Kalman variational auto-encoder* (KVAE) is based on the notion described above. To disentangle recognition and spatial representation, a sensory input $\mathbf{x}_t$ is mapped to $\mathbf{a}_t$ (VAE), a variable on a low-dimensional manifold that encodes an object's position and other visual properties. In turn, $\mathbf{a}_t$ is used as a pseudo-observation for the dynamics model (LGSSM). $\mathbf{x}_t$ represents a frame of a video[2] $\mathbf{x} = [\mathbf{x}_1, .., \mathbf{x}_T]$ of length $T$. Each frame is encoded into a point $\mathbf{a}_t$ on a low-dimensional manifold, so that the KVAE contains $T$ separate VAEs that share the same decoder $p_\theta(\mathbf{x}_t|\mathbf{a}_t)$ and encoder $q_\phi(\mathbf{a}_t|\mathbf{x}_t)$, and depend on each other through a time-dependent prior over $\mathbf{a} = [\mathbf{a}_1, .., \mathbf{a}_T]$. This is illustrated in figure 1.

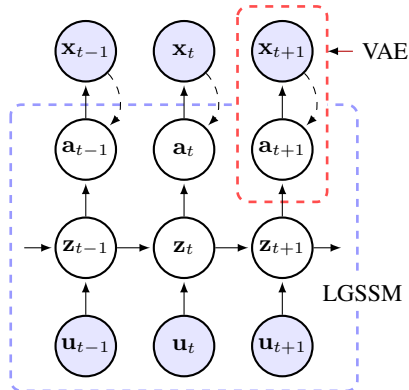

Figure 1: A KVAE is formed by stacking a LGSSM (dashed blue), and a VAE (dashed red). Shaded nodes denote observed variables. Solid arrows represent the generative model (with parameters $\theta$) while dashed arrows represent the VAE inference network (with parameters $\phi$).

### 3.1  Generative model

We assume that $\mathbf{a}$ acts as a latent representation of the whole video, so that the generative model of a sequence factorizes as $p_\theta(\mathbf{x}|\mathbf{a}) = \prod_{t=1}^{T} p_\theta(\mathbf{x}_t|\mathbf{a}_t)$. In this paper $p_\theta(\mathbf{x}_t|\mathbf{a}_t)$ is a deep neural network parameterized by $\theta$,

that emits either a factorized Gaussian or Bernoulli probability vector depending on the data type of $\mathbf{x}_t$. We model $\mathbf{a}$ with a LGSSM, and following (2), its prior distribution is

$$p_\gamma(\mathbf{a}|\mathbf{u}) = \int p_\gamma(\mathbf{a}|\mathbf{z})\, p_\gamma(\mathbf{z}|\mathbf{u})\, \mathrm{d}\mathbf{z}\ , \tag{3}$$

so that the joint density for the KVAE factorizes as $p(\mathbf{x}, \mathbf{a}, \mathbf{z}|\mathbf{u}) = p_\theta(\mathbf{x}|\mathbf{a})\, p_\gamma(\mathbf{a}|\mathbf{z})\, p_\gamma(\mathbf{z}|\mathbf{u})$. A LGSSM forms a convenient back-bone to a model, as the filtered and smoothed distributions $p_\gamma(\mathbf{z}_t|\mathbf{a}_{1:t}, \mathbf{u}_{1:t})$ and $p_\gamma(\mathbf{z}_t|\mathbf{a}, \mathbf{u})$ can be obtained exactly. Temporal reasoning can be done in the latent space of $\mathbf{z}_t$'s and via the latent encodings $\mathbf{a}$, and we can do long-term predictions without having to auto-regressively generate high-dimensional images $\mathbf{x}_t$. Given a few frames, and hence their encodings, one could "remain in latent space" and use the smoothed distributions to impute missing frames. Another advantage of using $\mathbf{a}$ to separate the dynamics model from $\mathbf{x}$ can be seen by considering the emission matrix $\mathbf{C}_t$. Inference in the LGSSM requires matrix inverses, and using it as a model for the prior dynamics of $\mathbf{a}_t$ allows the size of $\mathbf{C}_t$ to remain small, and not scale with the number of pixels in $\mathbf{x}_t$. While the LGSSM's process and measurement noise in (1) are typically formulated with full covariance matrices [26], we will consider them as isotropic in a KVAE, as $\mathbf{a}_t$ act as a prior in a generative model that includes these extra degrees of freedom.

What happens when a ball bounces against a wall, and the dynamics on $\mathbf{a}_t$ are not linear any more? Can we still retain a LGSSM backbone? We will incorporate nonlinearities into the LGSSM by regulating $\gamma_t$ from *outside* the exact forward-backward inference chain. We revisit this central idea at length in section 3.3.

## 3.2 Learning and inference for the KVAE

We learn $\theta$ and $\gamma$ from a set of example sequences $\{\mathbf{x}^{(n)}\}$ by maximizing the sum of their respective log likelihoods $\mathcal{L} = \sum_n \log p_{\theta\gamma}(\mathbf{x}^{(n)}|\mathbf{u}^{(n)})$ as a function of $\theta$ and $\gamma$. For simplicity in the exposition we restrict our discussion below to one sequence, and omit the sequence index $n$. The log likelihood or evidence is an intractable average over all plausible settings of $\mathbf{a}$ and $\mathbf{z}$, and exists as the denominator in Bayes' theorem when inferring the posterior $p(\mathbf{a}, \mathbf{z}|\mathbf{x}, \mathbf{u})$. A more tractable approach to both learning and inference is to introduce a variational distribution $q(\mathbf{a}, \mathbf{z}|\mathbf{x}, \mathbf{u})$ that approximates the posterior. The evidence lower bound (ELBO) $\mathcal{F}$ is

$$\log p(\mathbf{x}|\mathbf{u}) = \log \int p(\mathbf{x}, \mathbf{a}, \mathbf{z}|\mathbf{u}) \geq \mathbb{E}_{q(\mathbf{a},\mathbf{z}|\mathbf{x},\mathbf{u})} \left[ \log \frac{p_\theta(\mathbf{x}|\mathbf{a})p_\gamma(\mathbf{a}|\mathbf{z})p_\gamma(\mathbf{z}|\mathbf{u})}{q(\mathbf{a}, \mathbf{z}|\mathbf{x}, \mathbf{u})} \right] = \mathcal{F}(\theta, \gamma, \phi)\ , \tag{4}$$

and a sum of $\mathcal{F}$'s is maximized instead of a sum of log likelihoods. The variational distribution $q$ depends on $\phi$, but for the bound to be tight we should specify $q$ to be equal to the posterior distribution that only depends on $\theta$ and $\gamma$. Towards this aim we structure $q$ so that it incorporates the exact conditional posterior $p_\gamma(\mathbf{z}|\mathbf{a}, \mathbf{u})$, that we obtain with Kalman smoothing, as a factor of $\gamma$:

$$q(\mathbf{a}, \mathbf{z}|\mathbf{x}, \mathbf{u}) = q_\phi(\mathbf{a}|\mathbf{x})\, p_\gamma(\mathbf{z}|\mathbf{a}, \mathbf{u}) = \prod_{t=1}^T q_\phi(\mathbf{a}_t|\mathbf{x}_t)\, p_\gamma(\mathbf{z}|\mathbf{a}, \mathbf{u})\ . \tag{5}$$

The benefit of the LGSSM backbone is now apparent. We use a "recognition model" to encode each $\mathbf{x}_t$ using a non-linear function, after which exact smoothing is possible. In this paper $q_\phi(\mathbf{a}_t|\mathbf{x}_t)$ is a deep neural network that maps $\mathbf{x}_t$ to the mean and the diagonal covariance of a Gaussian distribution. As explained in section 4, this factorization allows us to deal with missing data in a principled way. Using (5), the ELBO in (4) becomes

$$\mathcal{F}(\theta, \gamma, \phi) = \mathbb{E}_{q_\phi(\mathbf{a}|\mathbf{x})} \left[ \log \frac{p_\theta(\mathbf{x}|\mathbf{a})}{q_\phi(\mathbf{a}|\mathbf{x})} + \mathbb{E}_{p_\gamma(\mathbf{z}|\mathbf{a},\mathbf{u})} \left[ \log \frac{p_\gamma(\mathbf{a}|\mathbf{z})p_\gamma(\mathbf{z}|\mathbf{u})}{p_\gamma(\mathbf{z}|\mathbf{a}, \mathbf{u})} \right] \right]\ . \tag{6}$$

The lower bound in (6) can be estimated using Monte Carlo integration with samples $\{\widetilde{\mathbf{a}}^{(i)}, \widetilde{\mathbf{z}}^{(i)}\}_{i=1}^I$ drawn from $q$,

$$\hat{\mathcal{F}}(\theta, \gamma, \phi) = \frac{1}{I} \sum_i \log p_\theta(\mathbf{x}|\widetilde{\mathbf{a}}^{(i)}) + \log p_\gamma(\widetilde{\mathbf{a}}^{(i)}, \widetilde{\mathbf{z}}^{(i)}|\mathbf{u}) - \log q_\phi(\widetilde{\mathbf{a}}^{(i)}|\mathbf{x}) - \log p_\gamma(\widetilde{\mathbf{z}}^{(i)}|\widetilde{\mathbf{a}}^{(i)}, \mathbf{u})\ . \tag{7}$$

Note that the ratio $p_\gamma(\widetilde{\mathbf{a}}^{(i)}, \widetilde{\mathbf{z}}^{(i)}|\mathbf{u}) / p_\gamma(\widetilde{\mathbf{z}}^{(i)}|\widetilde{\mathbf{a}}^{(i)}, \mathbf{u})$ in (7) gives $p_\gamma(\widetilde{\mathbf{a}}^{(i)}|\mathbf{u})$, but the formulation with $\{\widetilde{\mathbf{z}}^{(i)}\}$ allows stochastic gradients on $\gamma$ to also be computed. A sample from $q$ can be obtained by first sampling $\widetilde{\mathbf{a}} \sim q_\phi(\mathbf{a}|\mathbf{x})$, and using $\widetilde{\mathbf{a}}$ as an observation for the LGSSM. The posterior $p_\gamma(\mathbf{z}|\widetilde{\mathbf{a}}, \mathbf{u})$ can be tractably obtained with a Kalman smoother, and a sample $\widetilde{\mathbf{z}} \sim p_\gamma(\mathbf{z}|\widetilde{\mathbf{a}}, \mathbf{u})$ obtained from it. Parameter learning is done by *jointly* updating $\theta$, $\phi$, and $\gamma$ by maximising the ELBO on $\mathcal{L}$, which decomposes as a sum of ELBOs in (6), using stochastic gradient ascent and a single sample to approximate the intractable expectations.

### 3.3 Dynamics parameter network

The LGSSM provides a tractable way to structure $p_\gamma(\mathbf{z}|\mathbf{a}, \mathbf{u})$ into the variational approximation in (5). However, even in the simple case of a ball bouncing against a wall, the dynamics on $\mathbf{a}_t$ are not linear anymore. We can deal with these situations while preserving the linear dependency between consecutive states in the LGSSM, by non-linearly changing the parameters $\gamma_t$ of the model over time as a function of the latent encodings up to time $t-1$ (so that we can still define a generative model). Smoothing is still possible as the state transition matrix $\mathbf{A}_t$ and others in $\gamma_t$ do not have to be constant in order to obtain the exact posterior $p_\gamma(\mathbf{z}_t|\mathbf{a}, \mathbf{u})$.

Recall that $\gamma_t$ describes how the latent state $\mathbf{z}_{t-1}$ changes from time $t-1$ to time $t$. In the more general setting, the changes in dynamics at time $t$ may depend on the history of the system, encoded in $\mathbf{a}_{1:t-1}$ and possibly a starting code $\mathbf{a}_0$ that can be learned from data. If, for instance, we see the ball colliding with a wall at time $t-1$, then we know that it will bounce at time $t$ and change direction. We then let $\gamma_t$ be a learnable function of $\mathbf{a}_{0:t-1}$, so that the prior in (2) becomes

$$p_\gamma(\mathbf{a}, \mathbf{z}|\mathbf{u}) = \prod_{t=1}^{T} p_{\gamma_t(\mathbf{a}_{0:t-1})}(\mathbf{a}_t|\mathbf{z}_t) \cdot p(\mathbf{z}_1) \prod_{t=2}^{T} p_{\gamma_t(\mathbf{a}_{0:t-1})}(\mathbf{z}_t|\mathbf{z}_{t-1}, \mathbf{u}_t) . \qquad (8)$$

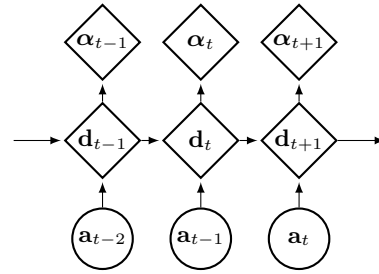

Figure 2: Dynamics parameter network for the KVAE.

During inference, after all the frames are encoded in $\mathbf{a}$, the dynamics parameter network returns $\gamma = \gamma(\mathbf{a})$, the parameters of the LGSSM at all time steps. We can now use the Kalman smoothing algorithm to find the exact conditional posterior over $\mathbf{z}$, that will be used when computing the gradients of the ELBO.

In our experiments the dependence of $\gamma_t$ on $\mathbf{a}_{0:t-1}$ is modulated by a *dynamics parameter network* $\boldsymbol{\alpha}_t = \boldsymbol{\alpha}_t(\mathbf{a}_{0:t-1})$, that is implemented with a recurrent neural network with LSTM cells that takes at each time step the encoded state as input and recurses $\mathbf{d}_t = LSTM(\mathbf{a}_{t-1}, \mathbf{d}_{t-1})$ and $\boldsymbol{\alpha}_t = \mathrm{softmax}(\mathbf{d}_t)$, as illustrated in figure 2. The output of the dynamics parameter network is weights that sum to one, $\sum_{k=1}^{K} \alpha_t^{(k)}(\mathbf{a}_{0:t-1}) = 1$. These weights choose and interpolate between $K$ different operating modes:

$$\mathbf{A}_t = \sum_{k=1}^{K} \alpha_t^{(k)}(\mathbf{a}_{0:t-1})\mathbf{A}^{(k)}, \quad \mathbf{B}_t = \sum_{k=1}^{K} \alpha_t^{(k)}(\mathbf{a}_{0:t-1})\mathbf{B}^{(k)}, \quad \mathbf{C}_t = \sum_{k=1}^{K} \alpha_t^{(k)}(\mathbf{a}_{0:t-1})\mathbf{C}^{(k)} . \quad (9)$$

We globally learn $K$ basic state transition, control and emission matrices $\mathbf{A}^{(k)}$, $\mathbf{B}^{(k)}$ and $\mathbf{C}^{(k)}$, and interpolate them based on information from the VAE encodings. The weighted sum can be interpreted as a soft mixture of $K$ different LGSSMs whose time-invariant matrices are combined using the time-varying weights $\boldsymbol{\alpha}_t$. In practice, each of the $K$ sets $\{\mathbf{A}^{(k)}, \mathbf{B}^{(k)}, \mathbf{C}^{(k)}\}$ models different dynamics, that will dominate when the corresponding $\alpha_t^{(k)}$ is high. The dynamics parameter network resembles the locally-linear transitions of [16, 33]; see section 6 for an in depth discussion on the differences.

## 4 Missing data imputation

Let $\mathbf{x}_{\mathrm{obs}}$ be an observed subset of frames in a video sequence, for instance depicting the initial movement and final positions of a ball in a scene. From its start and end, can we imagine how the ball reaches its final position? Autoregressive models like recurrent neural networks can only forward-generate $\mathbf{x}_t$ frame by frame, and cannot make use of the information coming from the final frames in the sequence. To impute the unobserved frames $\mathbf{x}_{\mathrm{un}}$ in the middle of the sequence, we need to do inference, not prediction.

The KVAE exploits the smoothing abilities of its LGSSM to use both the information from the past and the future when imputing missing data. In general, if $\mathbf{x} = \{\mathbf{x}_{\mathrm{obs}}, \mathbf{x}_{\mathrm{un}}\}$, the unobserved frames in $\mathbf{x}_{\mathrm{un}}$ could also appear at non-contiguous time steps, e.g. missing at random. Data can be imputed by sampling from the joint density $p(\mathbf{a}_{\mathrm{un}}, \mathbf{a}_{\mathrm{obs}}, \mathbf{z}|\mathbf{x}_{\mathrm{obs}}, \mathbf{u})$, and then generating $\mathbf{x}_{\mathrm{un}}$ from $\mathbf{a}_{\mathrm{un}}$. We factorize this distribution as

$$p(\mathbf{a}_{\mathrm{un}}, \mathbf{a}_{\mathrm{obs}}, \mathbf{z}|\mathbf{x}_{\mathrm{obs}}, \mathbf{u}) = p_\gamma(\mathbf{a}_{\mathrm{un}}|\mathbf{z}) \, p_\gamma(\mathbf{z}|\mathbf{a}_{\mathrm{obs}}, \mathbf{u}) \, p(\mathbf{a}_{\mathrm{obs}}|\mathbf{x}_{\mathrm{obs}}) , \qquad (10)$$

and we sample from it with ancestral sampling starting from $\mathbf{x}_{\mathsf{obs}}$. Reading (10) from right to left, a sample from $p(\mathbf{a}_{\mathsf{obs}}|\mathbf{x}_{\mathsf{obs}})$ can be approximated with the variational distribution $q_\phi(\mathbf{a}_{\mathsf{obs}}|\mathbf{x}_{\mathsf{obs}})$. Then, if $\gamma$ is fully known, $p_\gamma(\mathbf{z}|\mathbf{a}_{\mathsf{obs}}, \mathbf{u})$ is computed with an extension to the Kalman smoothing algorithm to sequences with missing data, after which samples from $p_\gamma(\mathbf{a}_{\mathsf{un}}|\mathbf{z})$ could be readily drawn.

However, when doing missing data imputation the parameters $\gamma$ of the LGSSM are not known at all time steps. In the KVAE, each $\gamma_t$ depends on all the previous encoded states, including $\mathbf{a}_{\mathsf{un}}$, and these need to be estimated before $\gamma$ can be computed. In this paper we recursively estimate $\gamma$ in the following way. Assume that $\mathbf{x}_{1:t-1}$ is known, but not $\mathbf{x}_t$. We sample $\mathbf{a}_{1:t-1}$ from $q_\phi(\mathbf{a}_{1:t-1}|\mathbf{x}_{1:t-1})$ using the VAE, and use it to compute $\gamma_{1:t}$. The computation of $\gamma_{t+1}$ depends on $\mathbf{a}_t$, which is missing, and an estimate $\widehat{\mathbf{a}}_t$ will be used. Such an estimate can be arrived at in two steps. The filtered posterior distribution $p_\gamma(\mathbf{z}_{t-1}|\mathbf{a}_{1:t-1}, \mathbf{u}_{1:t-1})$ can be computed as it depends only on $\gamma_{1:t-1}$, and from it, we sample

$$\widehat{\mathbf{z}}_t \sim p_\gamma(\mathbf{z}_t|\mathbf{a}_{1:t-1}, \mathbf{u}_{1:t}) = \int p_{\gamma_t}(\mathbf{z}_t|\mathbf{z}_{t-1}, \mathbf{u}_t)\, p_\gamma(\mathbf{z}_{t-1}|\mathbf{a}_{1:t-1}, \mathbf{u}_{1:t-1})\, \mathrm{d}\mathbf{z}_{t-1} \qquad (11)$$

and sample $\widehat{\mathbf{a}}_t$ from the predictive distribution of $\mathbf{a}_t$,

$$\widehat{\mathbf{a}}_t \sim p_\gamma(\mathbf{a}_t|\mathbf{a}_{1:t-1}, \mathbf{u}_{1:t}) = \int p_{\gamma_t}(\mathbf{a}_t|\mathbf{z}_t)\, p_\gamma(\mathbf{z}_t|\mathbf{a}_{1:t-1}, \mathbf{u}_{1:t})\, \mathrm{d}\mathbf{z}_t \approx p_{\gamma_t}(\mathbf{a}_t|\widehat{\mathbf{z}}_t)\,. \qquad (12)$$

The parameters of the LGSSM at time $t+1$ are then estimated as $\gamma_{t+1}([\mathbf{a}_{0:t-1}, \widehat{\mathbf{a}}_t])$. The same procedure is repeated at the next time step if $\mathbf{x}_{t+1}$ is missing, otherwise $\mathbf{a}_{t+1}$ is drawn from the VAE. After the forward pass through the sequence, where we estimate $\gamma$ and compute the filtered posterior for $\mathbf{z}$, the Kalman smoother's backwards pass computes the smoothed posterior. While the smoothed posterior distribution is not exact, as it relies on the estimate of $\gamma$ obtained during the forward pass, it improves data imputation by using information coming from the whole sequence; see section 5 for an experimental illustration.

## 5 Experiments

We motivated the KVAE with an example of a bouncing ball, and use it here to demonstrate the model's ability to separately learn a recognition and dynamics model from video, and use it to impute missing data. To draw a comparison with deep variational Bayes filters (DVBFs) [16], we apply the KVAE to [16]'s pendulum example. We further apply the model to a number of environments with different properties to demonstrate its generalizability. All models are trained end-to-end with stochastic gradient descent. Using the control input $\mathbf{u}_t$ in (1) we can inform the model of known quantities such as external forces, as will be done in the pendulum experiment. In all the other experiments, we omit such information and train the models fully unsupervised from the videos only. Further implementation details can be found in the supplementary material (appendix A) and in the Tensorflow [1] code released at github.com/simonkamronn/kvae.

### 5.1 Bouncing ball

We simulate 5000 sequences of 20 time steps each of a ball moving in a two-dimensional box, where each video frame is a 32x32 binary image. A video sequence is visualised as a single image in figure 4d, with the ball's darkening color reflecting the incremental frame index. In this set-up the initial position and velocity are randomly sampled. No forces are applied to the ball, except for the fully elastic collisions with the walls. The minimum number of latent dimensions that the KVAE requires to model the ball's dynamics are $\mathbf{a}_t \in \mathbb{R}^2$ and $\mathbf{z}_t \in \mathbb{R}^4$, as at the very least the ball's position in the box's 2d plane has to be encoded in $\mathbf{a}_t$, and $\mathbf{z}_t$ has to encode the ball's position and velocity. The model's flexibility increases with more latent dimensions, but we choose these settings for the sake of interpretable visualisations. The dynamics parameter network uses $K = 3$ to interpolate three modes, a constant velocity, and two non-linear interactions with the horizontal and vertical walls.

We compare the generation and imputation performance of the KVAE with two recurrent neural network (RNN) models that are based on the same auto-encoding (AE) architecture as the KVAE and are modifications of methods from the literature to be better suited to the bouncing ball experiments.[3]

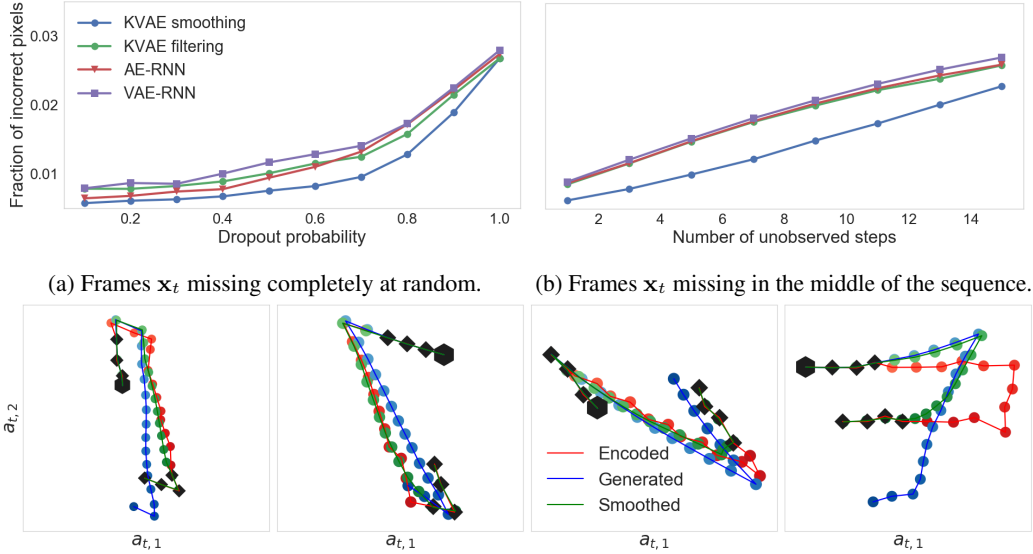

(a) Frames $\mathbf{x}_t$ missing completely at random.

(b) Frames $\mathbf{x}_t$ missing in the middle of the sequence.

(c) Comparison of encoded (ground truth), generated and smoothed trajectories of a KVAE in the latent space **a**. The black squares illustrate observed samples and the hexagons indicate the initial state. Notice that the $\mathbf{a}_t$'s lie on a manifold that can be rotated and stretched to align with the frames of the video.

Figure 3: Missing data imputation results.

In the *AE-RNN*, inspired by the architecture from [29], a pretrained convolutional auto-encoder, identical to the one used for the KVAE, feeds the encodings to an LSTM network [13]. During training the LSTM predicts the next encoding in the sequence and during generation we use the previous output as input to the current step. For data imputation the LSTM either receives the previous output or, if available, the encoding of the observed frame (similarly to filtering in the KVAE). The *VAE-RNN* is identical to the AE-RNN except that uses a VAE instead of an AE, similarly to the model from [6].

**Figure 3a** shows how well missing frames are imputed in terms of the average fraction of incorrectly guessed pixels. In it, the first 4 frames are observed (to initialize the models) after which the next 16 frames are dropped at random with varying probabilities. We then impute the missing frames by doing filtering and smoothing with the KVAE. We see in figure 3a that it is beneficial to utilize information from the whole sequence (even the future observed frames), and a KVAE with smoothing outperforms all competing methods. Notice that dropout probability 1 corresponds to pure generation from the models. **Figure 3b** repeats this experiment, but makes it more challenging by removing an increasing number of *consecutive* frames from the middle of the sequence ($T = 20$). In this case the ability to encode information coming from the future into the posterior distribution is highly beneficial, and smoothing imputes frames much better than the other methods. **Figure 3c** graphically illustrates figure 3b. We plot three trajectories over $\mathbf{a}_t$-encodings. The *generated* trajectories were obtained after initializing the KVAE model with 4 initial frames, while the *smoothed* trajectories also incorporated encodings from the last 4 frames of the sequence. The *encoded* trajectories were obtained with no missing data, and are therefore considered as ground truth. In the first three plots in figure 3c, we see that the backwards recursion of the Kalman smoother corrects the trajectory obtained with generation in the forward pass. However, in the fourth plot, the poor trajectory that is obtained during the forward generation step, makes smoothing unable to follow the ground truth.

The smoothing capabilities of KVAEs make it also possible to train it with up to 40% of missing data with minor losses in performance (appendix C in the supplementary material). Links to videos of the imputation results and long-term generation from the models can be found in appendix B and at sites.google.com/view/kvae.

**Understanding the dynamics parameter network.** In our experiments the dynamics parameter network $\boldsymbol{\alpha}_t = \boldsymbol{\alpha}_t(\mathbf{a}_{0:t-1})$ is an LSTM network, but we could also parameterize it with any differentiable function of $\mathbf{a}_{0:t-1}$ (see appendix D in the supplementary material for a comparison of various

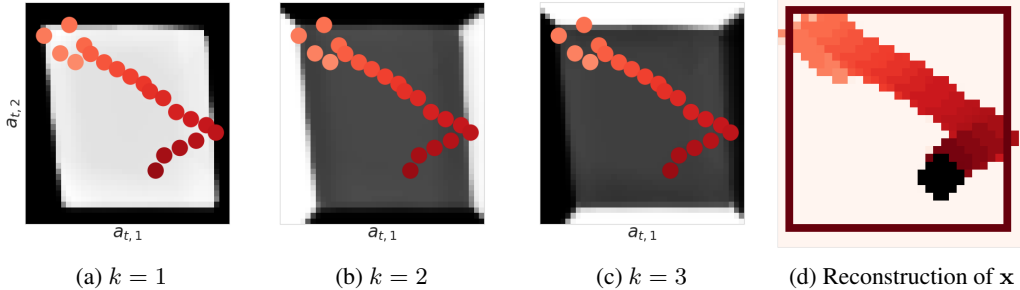

| (a) $k = 1$ | (b) $k = 2$ | (c) $k = 3$ | (d) Reconstruction of $\mathbf{x}$ |

Figure 4: A visualisation of the dynamics parameter network $\alpha_t^{(k)}(\mathbf{a}_{t-1})$ for $K = 3$, as a function of $\mathbf{a}_{t-1}$. The three $\alpha_t^{(k)}$'s sum to one at every point in the encoded space. The greyscale backgrounds in **a)** to **c)** correspond to the intensity of the weights $\alpha_t^{(k)}$, with white indicating a weight of one in the dynamics parameter network's output. Overlaid on them is the full latent encoding $\mathbf{a}$. **d)** shows the reconstructed frames of the video as one image.

architectures). When using a multi-layer perceptron (MLP) that depends on the previous encoding as mixture network, i.e. $\boldsymbol{\alpha}_t = \boldsymbol{\alpha}_t(\mathbf{a}_{t-1})$, figure 4 illustrates how the network chooses the mixture of learned dynamics. We see that the model has correctly learned to choose a transition that maintains a constant velocity in the center ($k = 1$), reverses the horizontal velocity when in proximity of the left and right wall ($k = 2$), the reverses the vertical velocity when close to the top and bottom ($k = 3$).

## 5.2   Pendulum experiment

We test the KVAE on the experiment of a dynamic torque-controlled pendulum used in [16]. Training, validation and test set are formed by 500 sequences of 15 frames of 16x16 pixels. We use a KVAE with $\mathbf{a}_t \in \mathbb{R}^2$, $\mathbf{z}_t \in \mathbb{R}^3$ and $K = 2$, and try two different encoder-decoder architectures for the VAE, one using a MLP and one using a convolutional neural network (CNN). We compare the performaces of the KVAE to DVBFs [16] and deep Markov models[4] (DMM) [19], non-linear SSMs parameterized by deep neural networks whose

| Model | Test ELBO |
|---|---|
| KVAE (CNN) | 810.08 |
| KVAE (MLP) | 807.02 |
| DVBF | 798.56 |
| DMM | 784.70 |

Table 1: Pendulum experiment.

intractable posterior distribution is approximated with an inference network. In table 1 we see that the KVAE outperforms both models in terms of ELBO on a test set, showing that for the task in hand it is preferable to use a model with simpler dynamics but exact posterior inference.

## 5.3   Other environments

To test how well the KVAE adapts to different environments, we trained it end-to-end on videos of (i) a ball bouncing between walls that form an irregular polygon, (ii) a ball bouncing in a box and subject to gravity, (iii) a Pong-like environment where the paddles follow the vertical position of the ball to make it stay in the frame at all times. Figure 5 shows that the KVAE learns the dynamics of all three environments, and generates realistic-looking trajectories. We repeat the imputation experiments of figures 3a and 3b for these environments in the supplementary material (appendix E), where we see that KVAEs outperform alternative models.

## 6   Related work

Recent progress in unsupervised learning of high dimensional sequences is found in a plethora of both deterministic and probabilistic generative models. The VAE framework is a common work-horse in the stable of probabilistic inference methods, and it is extended to the temporal setting by [2, 6, 8, 16, 19]. In particular, deep neural networks can parameterize the transition and emission distributions of different variants of deep state-space models [8, 16, 19]. In these extensions, inference

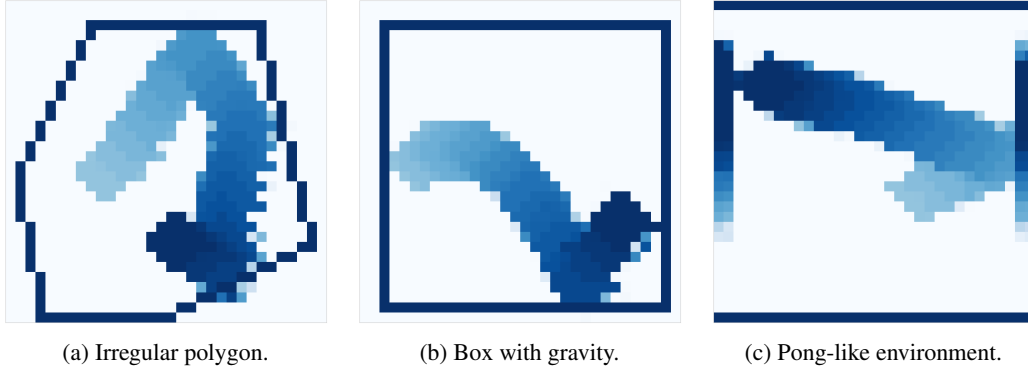

(a) Irregular polygon.          (b) Box with gravity.          (c) Pong-like environment.

Figure 5: Generations from the KVAE trained on different environments. The videos are shown as single images, with color intensity representing the incremental sequence index $t$. In the simulation that resembles Atari's Pong game, the movement of the two paddles (left and right) is also visible.

networks define a variational approximation to the intractable posterior distribution of the latent states at each time step. For the tasks in section 5, it is preferable to use the KVAE's simpler temporal model with an exact (conditional) posterior distribution than a highly non-linear model where the posterior needs to be approximated. A different combination of VAEs and probabilistic graphical models has been explored in [15], which defines a general class of models where inference is performed with message passing algorithms that use deep neural networks to map the observations to conjugate graphical model potentials.

In classical non-linear extensions of the LGSSM like the extended Kalman filter and in the locally-linear dynamics of [16, 33], the transition matrices at time $t$ have a non-linear dependence on $\mathbf{z}_{t-1}$. The KVAE's approach is different: by introducing the latent encodings $\mathbf{a}_t$ and making $\gamma_t$ depend on $\mathbf{a}_{1:t-1}$, the *linear* dependency between consecutive states of $\mathbf{z}$ is preserved, so that the exact smoothed posterior can be computed given $\mathbf{a}$, and used to perform missing data imputation. LGSSM with dynamic parameterization have been used for large-scale demand forecasting in [27]. [20] introduces recurrent switching linear dynamical systems, that combine deep learning techniques and switching Kalman filters [22] to model low-dimensional time series. [11] introduces a *discriminative* approach to estimate the low-dimensional state of a LGSSM from input images. The resulting model is reminiscent of a KVAE with no decoding step, and is therefore not suited for unsupervised learning and video generation. Recent work in the non-sequential setting has focused on disentangling basic visual concepts in an image [12]. [10] models neural activity by finding a non-linear embedding of a neural time series into a LGSSM.

Great strides have been made in the reinforcement learning community to model how environments evolve in response to action [5, 23, 24, 30, 32]. In similar spirit to this paper, [32] extracts a latent representation from a PCA representation of the frames where controls can be applied. [5] introduces action-conditional dynamics parameterized with LSTMs and, as for the KVAE, a computationally efficient procedure to make long term predictions without generating high dimensional images at each time step. As autoregressive models, [29] develops a sequence to sequence model of video representations that uses LSTMs to define both the encoder and the decoder. [7] develops an action-conditioned video prediction model of the motion of a robot arm using convolutional LSTMs that models the change in pixel values between two consecutive frames.

While the focus in this work is to define a generative model for *high dimensional* videos of simple physical systems, several recent works have combined physical models of the world with deep learning to learn the dynamics of objects in more complex but *low-dimensional* environments [3, 4, 9, 34].

## 7 Conclusion

The KVAE, a model for unsupervised learning of high-dimensional videos, was introduced in this paper. It disentangles an object's latent representation $\mathbf{a}_t$ from a latent state $\mathbf{z}_t$ that describes its dynamics, and can be learned end-to-end from raw video. Because the exact (conditional) smoothed posterior distribution over the states of the LGSSM can be computed, one generally sees a marked

improvement in inference and missing data imputation over methods that don't have this property. A desirable property of disentangling the two latent representations is that temporal reasoning, and possibly planning, could be done in the latent space. As a proof of concept, we have been deliberate in focussing our exposition to videos of static worlds that contain a few moving objects, and leave extensions of the model to real world videos or sequences coming from an agent exploring its environment to future work.

## Acknowledgements

We would like to thank Lars Kai Hansen for helpful discussions on the model design. Marco Fraccaro is supported by Microsoft Research through its PhD Scholarship Programme. We thank NVIDIA Corporation for the donation of TITAN X GPUs.

## Footnotes

[2]While our main focus in this paper are videos, the same ideas could be applied more in general to any sequence of high dimensional data.

[3]We also experimented with the SRNN model from [8] as it can do smoothing. However, the model is probably too complex for the task in hand, and we could not make it learn good dynamics.

[4]Deep Markov models were previously referred to as deep Kalman filters.

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
