[Supplementary Material · kvae_supplementary.pdf]

# A Disentangled Recognition and Nonlinear Dynamics Model for Unsupervised Learning
## *Supplementary material*

**Marco Fraccaro**[†*]   **Simon Kamronn** [†*]   **Ulrich Paquet**[‡]   **Ole Winther**[†]

[†] Technical University of Denmark

[‡] DeepMind

## A   Experimental details

We will describe here some of the most important experimental details. The rest of the details can be found in the code at github.com/simonkamronn/kvae.

**Data generation.**   All the videos were generated using the physics engine Pymunk. We generated 5000 videos for training and 1000 for testing.

**Encoder/Decoder architecture for the KVAE.**   As we only use image-based observations, the encoder is fixed to a three layer convolutional neural network with 32 units in each layer, kernel-size of 3x3, stride of 2, and ReLU activations. The decoder is an equally sized network using the Sub-Pixel[28] procedure for deconvolution. In the pendulum experiment however we also test MLPs.

**Optimization.**   As optimizer we use ADAM [17] with an initial learning rate of 0.007 and an exponential decay scheme with a rate of 0.85 every 20 epochs. Training one epoch takes 55 seconds on an NVIDIA Titan X and the model converges in roughly 80 epochs.

**Training tricks for end-to-end learning.**   The biggest challenge of this optimization problem is how to avoid poor local minima, for example where all the focus is given to the reconstruction term, at the expense of the prior dynamics given by the LGSSM. To achieve a quick convergence in all the experiments we found it helpful to

- downweight the reconstruction term from of VAEs during training, that is scaled by 0.3. By doing this, we can in fact help the model to focus on learning the temporal dynamics.
- learn for the first few epochs only the the VAE parameters $\theta$ and $\phi$ and the globally learned matrices $\mathbf{A}^{(k)}$, $\mathbf{B}^{(k)}$ and $\mathbf{C}^{(k)}$, but not the parameters of the dynamics parameter network $\boldsymbol{\alpha}_t(\mathbf{a}_{0:t-1})$. After this phase, all parameters are learned jointly. This allows the model to first learn good VAE embeddings and the scale of the prior, and then learn how to utilize the $K$ different dynamics.

**Choice of hyperparameters for the LGSSM.**   In most of the experiments we used $\mathbf{a}_t \in \mathbb{R}^2$, $\mathbf{z}_t \in \mathbb{R}^4$ and $K = 3$. In the *gravity* experiments we used however $\mathbf{z}_t \in \mathbb{R}^5$ as the model has no controls applied to it and needs to be able to learn a bias term due to the presence of the external force of gravity. The *polygon* experiments uses $K = 7$ as it needs to learn more complex dynamics. In general, we did not find difficult to tune the parameters of the KVAE, as the model can learn to prune unused components (if flexible enough).

---

[*]Equal contribution.

(a)                                      (b)

Figure 1: Training with missing data

## B   Videos

Videos are generated from all models by initializing with 4 frames and then sampling. The *filtering* and *smoothing* versions are allowed to observe part of the sequence depending on the masking scheme. All the *filtering* and *smoothing* videos are generated from sequences applied with a random mask with a masking probability of 80% (as in figure 3a) except for the videos with the suffix *consecutive* in which only the first and last 4 frames are observed (as in figure 3b). Only the KVAE models have *smoothing* videos. For the bouncing ball experiment (named *box* in the attached folder), we also show the videos from a model trained with 40% missing data.

In most videos the black ball is the ground truth, and the red is the one generated from the model, except for the ones marked *long_generation* in which the true sequence is not shown.

Videos are available from Google Drive and the website sites.google.com/view/kvae.

## C   Training with missing data.

The smoothed posterior described in section 4 can also be used to train the KVAE with missing data. In this case, we only need to modify the ELBO by masking the contribution of the missing data points in the joint probability distribution and variational approximation:

$$p(\mathbf{x}, \mathbf{a}, \mathbf{z}, \mathbf{u}) = p(\mathbf{z}_1) \prod_{t=2}^{T} p_{\gamma_t}(\mathbf{z}_t | \mathbf{z}_{t-1}, \mathbf{u}_t) \prod_{t=1}^{T} p_{\gamma_t}(\mathbf{a}_t | \mathbf{z}_t)^{\mathcal{I}_t} \prod_{t=1}^{T} p_\theta(\mathbf{x}_t | \mathbf{a}_t)^{\mathcal{I}_t}$$

$$q_\phi(\mathbf{a} | \mathbf{x}) = \prod_{t=1}^{T} q_\phi(\mathbf{a}_t | \mathbf{x}_t)^{\mathcal{I}_t} \ ,$$

where $\mathcal{I}_t$ is 0 if the data point is missing, 1 otherwise. Figure 6 illustrates a slight degradation in performance when training with respectively 30% and 40% missing data but, remarkably, the accuracy is still better when using smoothing in these conditions than with filtering with all training data available.

## D   Dynamics parameter network architecture

As the $\alpha$-network governs the non-linear dynamics, it has a significant impact on the modelling capabilities. Here we list the architectural choices considered:

- **MLP** with two hidden layers.
- **Recurrent Neural Networks** with LSTM units.
- **'First in, first out memory' (FIFO) MLP** with access to 5 time steps.

Figure 2: Comparison of modelling choices wrt. the $\alpha$-network

In all cases, we can also model $\alpha$ as an (approximate) discrete random variable using the the Concrete distribution [21, 14]. In this case we can recover an approximation to the switching Kalman filter[22].

In figure 7 the different choices are tested against each other on the bouncing ball data. In this case all the alternative choices result in poorer performances than the LSTM chosen for all the other experiments. We believe that LSTMs are able to better model the discretization errors coming from the collisions and the 32x32 rendering of the trajectories computed by the physics engine.

# E    Imputation in all environments

(a) Bouncing ball - Frames $\mathbf{x}_t$ missing randomly.

(b) Bouncing ball - Frames $\mathbf{x}_t$ missing in the middle

(c) Gravity - Frames $\mathbf{x}_t$ missing randomly.

(d) Gravity - Frames $\mathbf{x}_t$ missing in the middle

(e) Polygon - Frames $\mathbf{x}_t$ missing randomly.

(f) Polygon - Frames $\mathbf{x}_t$ missing in the middle

(g) Pong - Frames $\mathbf{x}_t$ missing randomly.

(h) Pong - Frames $\mathbf{x}_t$ missing in the middle

Figure 3: Imputation results for all environments