[Reviews · NeurIPS 2017]

Reviewer 1



This paper proposes to tackle disentanglement of videos into an object's representation and its dynamics, using a combination of Variational AutoEncoders and Kalman filters. The advantages of the method are: - The use of Linear Gaussian State Space Models allows for the exact inference of the posterior distribution on the spaces z - Encoding the observed video with VAE allows reasoning about the object's dynamics in low dimensional space - Finally, by making the LGSSM parameters non stationary, the state space transition can remain linear without loss of modelling ability The strengths of the paper are: - The paper is extremely clearly written, among the most well written the reviewer has read. - The problem to tackle and its challenge are clearly set - The proposed model gives better performance than compared models The weaknesses of the paper are mainly on the experiments: - While not familiar with the compared models DMM and DVBF in details, the reviewer understood from the paper their differences with KVAE. However, the reviewer would appreciate a little bit more detailed presentation of the compared models. Specifically, the KVAE is simpler as the state space transition are linear, but it requires the computation of the time-dependant LGSSM parameters \gamma. Can the authors comment on the computation requirements of the 3 methods compared in Table 1 ? - Why the authors did not test DMM and DVBF on the task of imputing missing data ?

Reviewer 2



The paper presents a time-series model for high dimensional data by combining variational auto-encoder (VAE) with linear Gaussian state space model (LGSSM). The proposed model takes the latent repressentation from VAE as the output of LGSSM. The exact inference of linear Gaussian state space model via Kalman smoothing enables efficient and accurate variational inference for the overall model. To extend the temporal dynamics beyond linear dependency, the authors use a LSTM to parameterize the matrices in LGSSM. The performance of the proposed model is evaluated through bouncing ball and Pendulum experiments. LGSSM with dynamic parameterization is a nice trick to learn complex temporal dynamics through a simple probabilistic model, but has been exploited in the literature, e.g., [M. Seeger, D. Salinas, V. Flunkert NIPS 2016]. The proposed method does not really disentangle the object representation from the temporal dynamic, as the model is unlikely to infer the correct dynamics of an unseen object. The proposed method only models the temporal dynamic on top of the low dimensional latent representation from VAE. The proposed method is evaluted on some simple examples, where the object representation is simple (mostly single object with low resolution image) and the dynamic is also simple (bouncing of a ball or Pendulum). It would be more convencing to show experiments with either complex data representation or complex dynamics (e.g., human motion). In Pendulum experiment, the proposed method is compared with other two methods in terms of ELBO. However, it is questionable how to compare ELBO from different models (e.g., the numbers of parameters are probably different.).

Reviewer 3



This paper seems like a natural combination of VAEs to model image generation and Kalman filters for modeling dynamics. It seems to fill a missing area in the space of models. Comments: The notation $p_{\theta\gamma}$ on L101 is confusing It would be more clear to include \theta, \gamma, and \phi in Figure 1, since otherwise it is easy to forget which is which It would be nice if Table 1 also included some confidence intervals Questions: How necessary is the dynamics parameter network in Sec 3.3? What happens if the nonlinearity is handled using traditional EKF or UKF techniques? Latex/Formatting: Appendix, Table, Figure should be capitalized see citep vs citet see \times for L204